# Exposure to landscape fire smoke reduced birthweight in low- and middle-income countries: findings from a siblings-matched case-control study

Jiajianghui Li[1†], Tianjia Guan[2†], Qian Guo[3†], Guannan Geng[4], Huiyu Wang[1], Fuyu Guo[1], Jiwei Li[5], Tao Xue[1]*

[1]Institute of Reproductive and Child Health / Ministry of Health Key Laboratory of Reproductive Health and Department of Epidemiology and Biostatistics, School of Public Health, Peking University Health Science Centre, Beijing, China; [2]Department of Health Policy, School of Health Policy and Management, Chinese Academy of Medical Sciences and Peking Union Medical College, Beijing, China; [3]School of Energy and Environmental Engineering, University of Science and Technology, Beijing, China; [4]School of Environment, Tsinghua University, Beijing, China; [5]College of Computer Science and Technology, Zhejiang University, Hangzhou, China

*For correspondence:
txue@hsc.pku.edu.cn

[†]These authors contributed equally to this work

Competing interest: The authors declare that no competing interests exist.

## Abstract

**Background:** Landscape fire smoke (LFS) has been associated with reduced birthweight, but evidence from low- and middle-income countries (LMICs) is rare.

**Methods:** Here, we present a sibling-matched case–control study of 227,948 newborns to identify an association between fire-sourced fine particulate matter ($PM_{2.5}$) and birthweight in 54 LMICs from 2000 to 2014. We selected mothers from the geocoded Demographic and Health Survey with at least two children and valid birthweight records. Newborns affiliated with the same mother were defined as a family group. Gestational exposure to LFS was assessed in each newborn using the concentration of fire-sourced $PM_{2.5}$. We determined the associations of the within-group variations in LFS exposure with birthweight differences between matched siblings using a fixed-effects regression model. Additionally, we analyzed the binary outcomes of low birthweight (LBW) or very low birthweight (VLBW).

**Results:** According to fully adjusted models, a 1 μg/m³ increase in the concentration of fire-sourced $PM_{2.5}$ was significantly associated with a 2.17 g (95% confidence interval [CI] 0.56–3.77) reduction in birthweight, a 2.80% (95% CI 0.97–4.66) increase in LBW risk, and an 11.68% (95% CI 3.59–20.40) increase in VLBW risk.

**Conclusions:** Our findings indicate that gestational exposure to LFS harms fetal health.

**Funding:** PKU-Baidu Fund, National Natural Science Foundation of China, Peking University Health Science Centre, and CAMS Innovation Fund for Medical Sciences.

## Introduction

The natural cycle of landscape fires (e.g., wildfires, tropical deforestation fires, and agricultural biomass burning) plays an important role in maintaining the terrestrial ecosystem. Yet, landscape fire smoke (LFS) triggers a costly and growing global public health problem (*van der Werf et al., 2010*). The

frequency and intensity of landscape fire events have tended to increase (*Cascio, 2018*; *Turetsky et al., 2010*), driven by interactions between human activities (e.g., slash and slave agriculture) and climate change (*Hantson et al., 2015*). Most emissions originate from fires located in tropical rainforests and savannas, where they cause recurrent episodes of severe ambient pollution that affect mainly low- and middle-income countries (LMICs) (*van der Werf et al., 2010*). Worldwide concern regarding large-scale natural forest burning has sparked increasing interest in the harmful effects of LFS exposure on human health.

LFS is composed of hundreds of combustion products, such as carbon monoxide, nitrogen oxides, and polycyclic aromatic hydrocarbons (*Matz et al., 2020*). Particulate matter with an aerodynamic diameter ≤2.5 µm ($PM_{2.5}$) is one of the most widely studied pollutants derived from LFS. The 24 hr average $PM_{2.5}$ concentrations on burning days can be many times higher than those on normal days, and the peak level of ambient exposure can persist for several weeks (*Barn et al., 2016*). Several epidemiological studies have shown adverse effects of LFS exposure on human health. The majority of the study outcomes involved morbidity and mortality caused by acute respiratory and cardiovascular diseases, particularly among vulnerable individuals, such as children and the elderly (*Delfino et al., 2009*; *Johnston et al., 2012*; *Morgan et al., 2010*). However, the health impacts of LFS on susceptible pregnant women, another highly vulnerable group due to gestation-related physiological changes, such as an increased breathing rate during pregnancy (*Kolarzyk et al., 2005*), are not thoroughly discussed. Previous studies have shown that gestational exposure to air pollutants, including those derived from LFS and other emissions, is associated with abnormal placental vascular function and, consequently, low birthweight (LBW) (*Pedersen et al., 2013*), preterm birth (*Padula et al., 2019*), and small for gestational age (*Bijnens et al., 2016*). Exposure to air pollution induces subclinical disorders, such as inflammation, oxidative stress, increased blood viscosity, mitochondrial methylation, and hypoxia (*Janssen et al., 2015*; *Lee et al., 2004*), which renders the association between LFS and adverse maternal outcomes biologically plausible.

Birth weight comprehensively reflects the intrauterine environment quality, fetal growth, and maternal nutritional status. Therefore, LBW is a global public health problem. LBW neonates are at higher risk for a range of diseases in later life (e.g., early growth retardation and adulthood cardiorespiratory diseases) compared with normal newborns (*Hviid and Melbye, 2007*; *Samaras et al., 2003*).

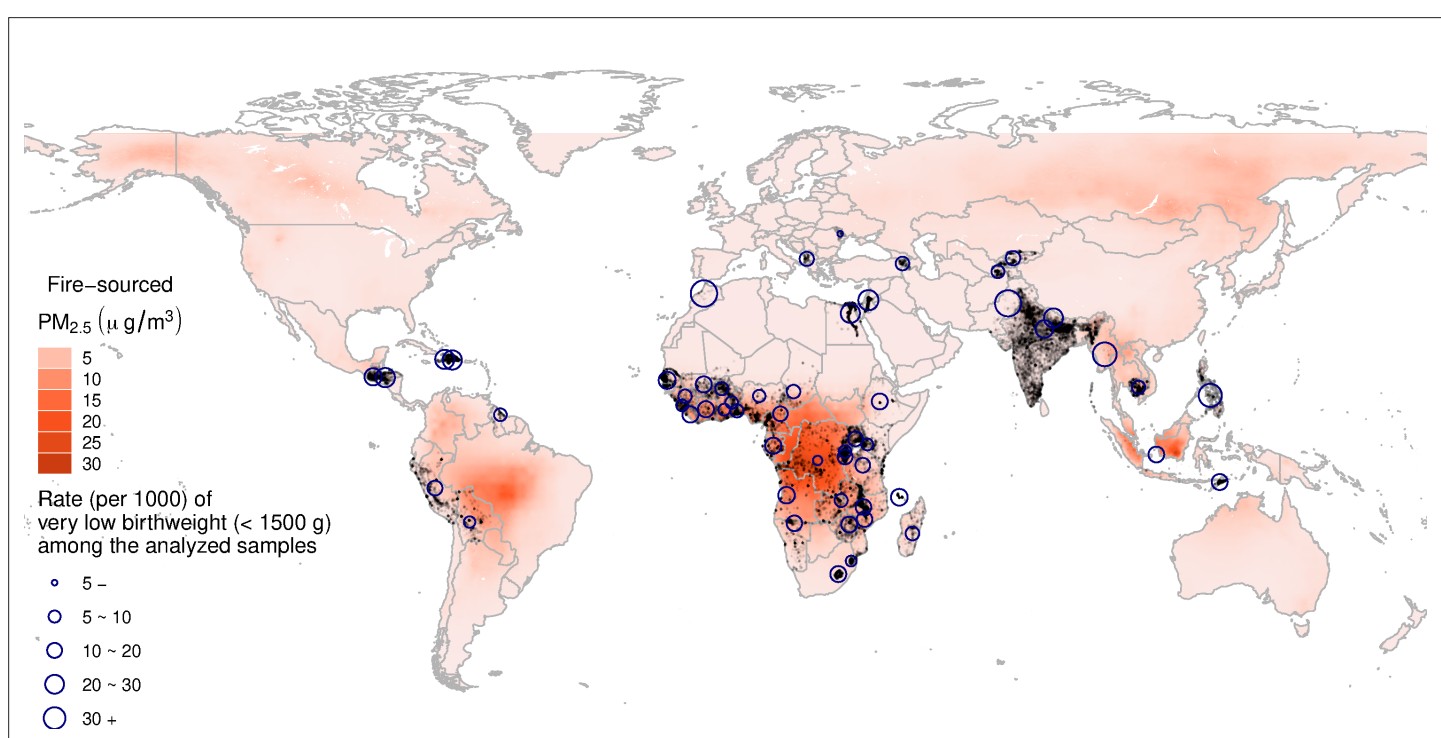

**Figure 1.** Distribution of the fire-sourced $PM_{2.5}$ concentrations (average for 2000–2014) and the locations of the analyzed livebirths (gray dots). The circles represent the rate of very low birthweight births among the analyzed livebirths by country.

The estimated worldwide prevalence of LBW was 14.6% (95% confidence interval [CI] 12.4–17.1) in 2015, compared with 17.5% (95% CI 14.1–21.3) in 2000 (*Blencowe et al., 2019*). Although some progress has been made in reducing LBW globally, the average annual reduction rate is insufficient to achieve the 2025 WHO global target (*World Health organization, 2017*). Additionally, the relevant burden of LBW is unequally distributed in the world as 91% of LBW infants are from LMICs, particularly from Southern Asia and Sub-Saharan Africa (*Blencowe et al., 2019*).

All risk factors for adverse birth outcomes should be screened thoroughly to protect maternal and infant health. In addition to conventional factors, such as nutrition and maternal condition, environmental factors, including ambient $PM_{2.5}$ exposure, also contribute to a considerable global burden of adverse birth outcomes (*Malley et al., 2017*). Although a growing number of studies have suggested the deleterious effects of urban air pollution on LBW (*Li et al., 2020*; *Li et al., 2017*), the exposure–response function specifically for fire-sourced $PM_{2.5}$ has been insufficiently studied. Furthermore, studies concerning the association between LFS and birthweight have focused primarily on high-income countries, and their conclusions have been inconsistent (*Abdo et al., 2019*; *Cândido da Silva et al., 2014*; *Holstius et al., 2012*; *O'Donnell and Behie, 2015*). As most landscape fires occur in LMICs (*Figure 1*), studies to assess the relevant health impacts of LFS among residents of such countries are warranted. Hence, we explored the association between birthweight and fire-sourced $PM_{2.5}$ in 54 LMICs using a sibling-matched case–control study from 2000 to 2014, which was based on a series of global surveys.

## Methods
### Study population
The birthweight data and relevant individual variables were extracted from 113 individual Demographic and Health Surveys (DHSs) representing 54 LMICs from 2000 to 2014 (*Figure 1*). The DHSs were conducted by the US Agency for International Development to characterize the demography and health of more than 90 developing countries. DHSs are usually conducted every 5 years in each country using a two-stage sampling design. Information on demographic factors, socioeconomic factors, and health status was collected from a nationally representative probability sample of households. Well-trained fieldworkers conducted the surveys using uniformly designed questionnaires in the native language. The primary sampling units (e.g., villages or blocks) have begun to be geocoded in recent waves using global positioning systems (GPSs). The GPS-enhanced DHSs were matched with environmental exposure and thus included in our analyses. Please refer to the open-access website for detailed information on the DHS database: https://www.dhsprogram.com/.

All females aged 15–49 years within each selected household were eligible respondents of the survey. The health outcome analyzed in this study was birthweight. The DHSs asked the screened mothers to recall the birthweight of all children born during the 5 years leading up to the survey date. To characterize different levels of reduced birthweight, we examined two binary outcomes: LBW (birthweight <2500 g) and, other covariates, such as maternal age and birth time (in months), were extracted from the relevant questionnaires.

### Exposure assessment
We assessed ambient exposure to LFS as the $PM_{2.5}$ concentration attributable to landscape fires (hereafter, defined as fire-sourced $PM_{2.5}$), referring to a previous study (*Stowell et al., 2019*). The fire-sourced $PM_{2.5}$ concentrations were estimated using the GEOS-Chem . The detailed settings of the GEOS-Chem are provided in the previous publication (*Xue et al., 2021b*) and briefly summarized here. We used the 11-01 version of GEOS-Chem model (https://geos-chem.seas.harvard.edu/). The meteorological fields to drive the model were directly obtained from Modern-Era Retrospective analysis for Research and Applications Version 2 (MERRA-2), which is freely distributed by the Goddard Earth Sciences Data and Information Services Center (https://disc.gsfc.nasa.gov/). Anthropogenic emissions inputted into the model were obtained from the global inventory of Community Emissions Data System (CEDS) (*Hoesly et al., 2018*), from 2000 to 2014; the fire emissions during the same period were collected from the Global Fire Emission Database (GFED4s, https://www.globalfiredata.org/). The GEOS-Chem finally simulated the ground-surface concentrations of $PM_{2.5}$ in a regular grid of 2° × 2.5° across the world.

The fire-sourced PM$_{2.5}$ was derived from a comparison between two GEOS-Chem model runs. The two models were identical except for switching on and off fire emissions. We quantified the fraction ($\rho_m$) of fire-attributed PM$_{2.5}$ using the following equation:

$$\rho_{my} = (\text{PM}^{on}_{2.5,my} - \text{PM}^{off}_{2.5,my})/\text{PM}^{om}_{2.5,my} \tag{1}$$

where the superscripts *on* and *off* denote the two simulations switching on and off fire emissions, respectively; and the subscripts *m* and *y* are month and year indicators, respectively.

Additionally, considering the potential errors in the CTM procedure (*Solazzo et al., 2012*), the bias correction approach has been widely applied to improve CTM performance in exposure assessments using ground-surface observational data (*Zhang et al., 2020*). We utilized a well-developed and widely utilized product of satellite-based PM$_{2.5}$ estimates (*van Donkelaar et al., 2016*) (PM$_{2.5, y}{}^{satellite}$) in the bias correction. This product has a spatial resolution of 0.05° × 0.05° and was publicly available from 1998 to 2018 as the gridded annual mean (https://sites.wustl.edu/acag/datasets/surface-pm2-5/#V4.GL.03). However, since a key input into the product, satellite measurements of aerosol from the Moderate Resolution Imaging Spectroradiometer are only available after February 2000, we believe that the estimates after the date have a better quality than those before. Therefore, we utilized the product to calibrate the monthly GEOS-Chem simulations from February 2000 to December 2014. We prepared the GEOS-Chem simulations ($\rho_{m,y}$ and PM$_{2.5, m,y}{}^{on}$) in the same 0.05° × 0.05° grid using an inversed distance-weighted downscaling approach before bias correction. The bias correction rate ($\eta_y$) was calculated as

$$\eta = \text{PM}^{satellite}_{2.5,y} \div (1/12 * \sum_m \text{PM}^{on}_{2.5,m,y}) \tag{2}$$

The monthly exposure indicator of fire-sourced or non-fire-sourced PM$_{2.5}$ was finally derived as

$$[\text{Fire} - \text{sourced PM}_{2.5}]_{m,y} = \eta_y \times \rho_{m,y} \times \text{PM}^{on}_{2.5,m,y},$$
$$[\text{Non} - \text{fire} - \text{sourced PM}_{2.5}]_{m,y} = \eta_y \times (1 - \rho_{m,y}) \times \text{PM}^{on}_{2.5,m,y} \tag{3}$$

We utilized the non-fire-sourced PM$_{2.5}$ as a potential confounder in the subsequent epidemiological analysis. We also collected other environmental variables, including ambient temperature and humidity data, from an assimilated dataset called Modern-Era Retrospective analysis for Research and Applications, version 2 (please refer to the supplemental text for more details). The variables were downloaded as monthly gridded values and prepared in the 0.05° × 0.05° grid in the same way as we processed the GEOS-Chem outputs.

Since fire-sourced PM$_{2.5}$ is mixed with particles from other sources, it cannot be easily measured. Although the CTM-based approach has been widely utilized in a few large-scale studies (*Xue et al., 2021a*; *Ye et al., 2021*) to assess the LFS exposure, accuracy of its results cannot be evaluated directly. Therefore, in this study, we assessed the overall performance of the GEOS-Chem model by comparing the simulated PM$_{2.5}$ concentrations with the satellite-based estimates (*van Donkelaar et al., 2016*). At the surveyed locations (i.e., the gray dots in *Figure 1*), we found that the two estimators were in good agreement with each other (Pearson correlation coefficient, $R^2$ = 0.76; more details on the comparison are shown in *Supplementary file 1*).

To understand the associations between birthweight and environmental variables, we focused on exposure during gestation. As the DHSs did not record the specific duration of gestation, we utilized the 9- month average preceding the birth month as exposure time during pregnancy.

## Study design

To evaluate the association between fire-sourced PM$_{2.5}$ and birthweight, we applied a sibling-matched case–control study, similar to our previous work (*Xue et al., 2021b*; *Xue et al., 2021c*; *Xue et al., 2021c*; *Xue et al., 2019*). We defined the descendants of the same mother as a family group of matched siblings and compared birthweights within each group. The family-level baseline birthweight (defined as the mean birthweight within each group) can be affected by many complex factors, including genetics, socioeconomic position, and quality of local medical services, which can be difficult to measure. We matched the mothers to control for those unmeasured confounders in the study design. A sibling-matched case–control design is a simple, robust, and cost-efficient method, particularly for large population data with potential heterogeneity (*Colley et al., 2019*; *Curtis, 1997*).

We extracted the females with at as least two valid records of newborns from all available individual DHSs. Each valid record was defined using the following inclusion criteria: (1) a valid birthweight, (2) a record of the GPS coordinates, (3) a valid birthdate, and (4) valid environmental exposure values (some of the satellite-based $PM_{2.5}$ estimates were not available for some locations, such as small islands). Finally, our dataset assembled 113 individual surveys from 54 countries (*Figure 1*) and included 227,948 newborns classified into 109,137 groups according to their mothers.

## Statistical analyses

According to the sibling-matched method, we used a fixed-effects regression model to evaluate the associations between fire-sourced $PM_{2.5}$ and LBW or VLBW. The model was specified as

$$y_{i,j} \sim x_{i,j}\beta + z_{i,j} + \Theta_i \qquad (4)$$

where the subscripts $i$ and $j$ denote the indices of the mother-affiliated group and child, respectively; $x_{i,j}$ denotes the concentration of fire-sourced $PM_{2.5}$; $z_{i,j}$ denotes the adjusted covariates; $y_{i,j}$ denotes the outcome variable; and $\theta_i$ is a nuisance parameter for the fixed effect to control for group-specific unmeasured confounders. The model was specified as Gaussian regression when setting $y_{i,j}$ as the birthweight and as logit regression when setting $y_{i,j}$ as the logit-transformed probability of LBW or VLBW. The adjusted covariates included maternal age; child sex; multiple births; non-fire-sourced $PM_{2.5}$; spline terms of birth order (with 5 degrees of freedom [DF]), of temperature (3 DF), of humidity (3 DF), of the calendar year (5 DF), and of the monthly index (4 DF); and a spatiotemporal effect parameterized by the interaction between country and calendar year. The nonlinear effects of month and year controlled for the seasonal periodic variation and long-term trend in child health, respectively. The spatiotemporal effect captured country-specific trends. The association between birthweight and LFS was evaluated by the regression coefficient $\beta$ for 1 µg/m³ increases in fire-sourced $PM_{2.5}$. The association with LBW or VLBW was evaluated by excess risk [ = (1 − exp($\beta$)) × 100%] for a 1 µg/m³ increase in fire-sourced $PM_{2.5}$. All environmental exposures were quantified using the same time window. The major time window was the 9 months before birth. We also calculated the alternative exposure indicator as an average during the 3 or 6 months preceding birth.

In sensitivity analyses, we first derived the lag-distributed model to explore how the association varied during the exposure time window. As the specific duration of gestation was not recorded, before actual data analysis, we utilized the 9- month average preceding the birth month as the major exposure time window. After conducting the major analysis, we re-estimated the effects of fire-sourced $PM_{2.5}$ on birthweight change, LBW, or VLBW, by different time windows using the lag-distributed model. Next, subpopulation-specified associations were estimated using interaction analyses to detect potential heterogeneity. Third, we developed nonlinear associations by replacing the linear term of fire-sourced $PM_{2.5}$ using a set of thin-plate spline functions. Fourth, to examine potential recall bias, we estimated the associations by strata of the durations from birth to survey time (the duration is defined as recall period in this study). We assumed a shorter recall period indicated for a lower likelihood of outcome misclassification. Fifth, to test whether the estimated associations were attributable to LFS exposure or other direct damage related to landscape fires (e.g., destroying human habitats), we derived an indicator for the transported fire-sourced $PM_{2.5}$, according to a satellite image of the burned area (MCD64A1, https://lpdaacsvc.cr.usgs.gov/appeears/). The fire-sourced $PM_{2.5}$ at the pixels with the zero-burned area during gestation were defined as smoke transported from other locations. The associations with transported fire-sourced $PM_{2.5}$ could be indicative of the effect of LFS exposure via the inhalation pathway.

Family-level baseline birthweight and the correlated gestation length can be determined by factors such as genetics and socioeconomic level, which may also affect the association between the within-group change in birthweight and LFS exposure. The fixed-effects regression models actually infer an association by examining the coherence between derivation of individual-level birthweight from the family-level mean (i.e., the expected birthweight given a specific family) and the corresponding exposure. To show that, we defined the family-level baseline birthweight as $y_i$. Equation (4) was equivalently

transformed as $\Delta y_{i,j} = (y_{i,j} - y_i) \sim x_{i,j}\,\beta + \mathbf{z}_{i,j}\boldsymbol{\gamma} + (\theta_i - y_i)$, where $(\theta_i - y_i)$ is viewed as a new nuisance parameter $\theta_i^*$. The nature of fixed-effects model made it possible to explore how the effect of fire-sourced PM$_{2.5}$ varied according to family-level mean birthweight. Accordingly, we derived a baseline-varying association model as follows:

$$\triangle y_{i,j} \sim f(y_i)x_{i,j} + z_{i,j}y + \Theta_i^* \tag{5}$$

All analyses were performed in R (version 4.0.2; The R Foundation for Statistical Computing, Vienna, Austria). Inference for the fixed-effects models was performed using the *fixest* package. The lag-distributed effect was parameterized using the *dlnm* package, and the thin-plate splines for the nonlinear association and baseline-varying effect were parameterized using the *mgcv* package. Source data for figures and R codes for the epidemiological models are documented in Source code 1.

## Results

### Descriptive summary

Among the 227,948 livebirths analyzed from 54 LMICs, there were 109,137 groups of siblings who were matched to their mothers. Each group had an average of 2.13 (standard deviation [SD] 0.36) livebirths. Approximately half of the livebirths were from Sub-Saharan Africa and one-quarter from South Asia. The mean birthweight was 3,082 (SD 724) g. There were 31,854 LBW births and 2,912 very low birthweight (VLBW) births. More statistics on the analyzed livebirths are shown in *Supplementary file 1b*.

The spatial distribution of the long-term average fire-sourced PM$_{2.5}$ concentrations is displayed in *Figure 1*. Our modeling results indicate that the hotspots of LFS include regions of the Congo rainforest basin, Amazon rainforest basin, Indonesian forests, Siberia forests, and other forest areas. Our analyzed samples included newborns from the first three regions. Among our samples, the average concentration of full gestational exposure to fire-sourced PM$_{2.5}$ was 4.29 (SD: 5.53) µg/m$^3$. A detailed summary of the environmental exposures is presented in *Supplementary file 1b*.

Gestational exposure to fire-sourced PM$_{2.5}$ varied widely between family groups; it was higher in those living closer to hotspot regions (e.g., forests). Approximately 92% of the variation in fire-sourced PM$_{2.5}$ exposure was between family groups and 8% was within family groups. For the total variation in birthweight, 71.7% was between groups and 28.3% within groups (*Table 1*). The within- or between-group variations reflected temporal or spatial changes in those variables, respectively. Birthweight and fire-sourced PM$_{2.5}$ were negatively correlated in the temporal dimension but positively correlated in the spatial dimension. As the spatial patterns in birthweight and fire-sourced PM$_{2.5}$ were determined by geographic, socioeconomic, and cultural factors, the between-group correlations may have been confounded by their common determiners, such as urbanization level. The complex correlations between birthweight and LFS suggest that statistical inference of their associations should be cautiously determined based on well-designed epidemiological analysis.

### Association between LFS and birthweight

Although the fully adjusted model indicated that a 1 µg/m$^3$ increase in exposure to fire-sourced PM$_{2.5}$ during the 9 months before birth was significantly associated with a 2.17 g (95% CI 0.56–3.77) birthweight reduction, the association was sensitive to differences in the adjusted covariates and exposure

**Table 1.** Between- or within-group variations in birthweight and gestational exposure to fire-sourced PM$_{2.5}$ and their correlations.

|  | Standard deviation (% of total variance) | | |
|---|---|---|---|
|  | **Birthweight(g)** | **Fire-sourced PM$_{2.5}$ g/m$^3$** | **Correlation (R)** |
| Total | 724 (100%) | 5.53 (100%) | 0.1594 |
| Between groups of matched siblings | 613 (71.7%) | 5.30 (91.7%) | 0.1973 |
| Within groups of matched siblings | 386 (28.3%) | 1.59 (8.3%) | −0.0035 |

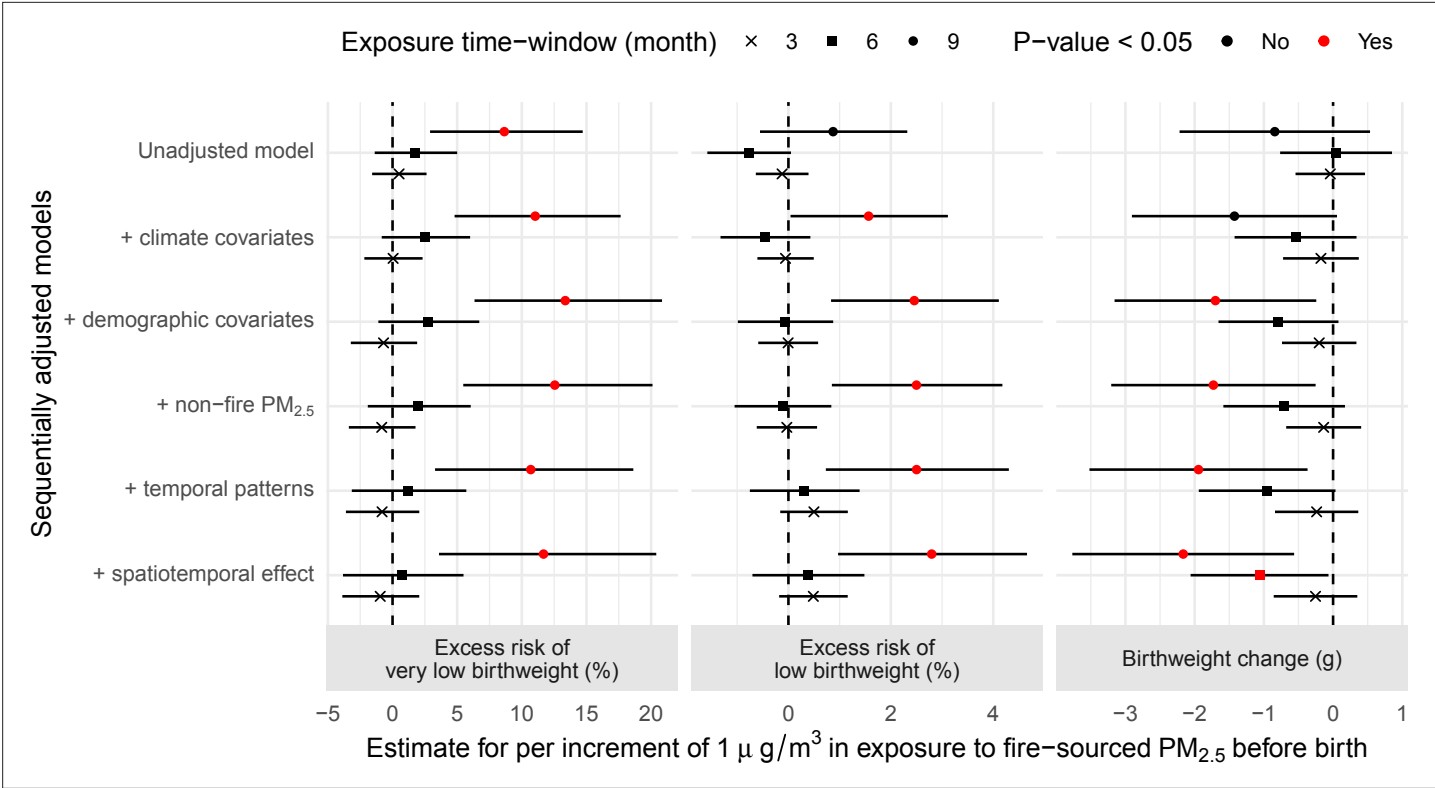

**Figure 2.** Effects of fire-sourced PM$_{2.5}$, estimated by linear models. The dots with error bars show the estimated associations between gestational exposure to fire-sourced PM$_{2.5}$ and birthweight change, low birthweight, or very low birthweight. The dots represent the point estimates, and bars represent the corresponding 95% confidence intervals.

The online version of this article includes the following figure supplement(s) for figure 2:

**Figure supplement 1.** The estimated associations between fire-sourced PM$_{2.5}$ and birthweight change, low birthweight or very low birthweight, by different lags.

**Figure supplement 2.** The cumulated birthweight change associated with an average of fire-sourced PM$_{2.5}$ concentrations from a lagged month to birth.

**Figure supplement 3.** The subpopulation-specific associations between gestational exposure to fire-sourced PM$_{2.5}$ and birthweight change, low birthweight and very low birthweight.

time window (*Figure 2*). For example, the unadjusted model reported the estimated effect as a birthweight reduction of 0.84 g (95% CI −0.53 to 2.22). The lag-specific associations did not display a clear time-varying pattern (*Figure 2—figure supplement 1*), which suggests a linear increase in strength in the association with increased length of the exposure time window (*Figure 2*, *Figure 2—figure supplement 2*). The subpopulation-specific results suggest that the association was potentially heterogeneous (*Figure 2—figure supplement 3*). The association was stronger for female infants, newborns of nulliparous mothers, or newborns of unemployed mothers compared with the corresponding references (i.e., male infants, newborns of multiparous mothers, or newborns of employed mothers, respectively). The nonlinear analysis confirmed the adverse effect of LFS on birthweight (*Figure 3*). The sublinear association suggests a saturated effect of an ~60 g birthweight reduction for exposure to extreme fires, which contributed to a PM$_{2.5}$ concentration >~20 µg/m$^3$.

## Association between LFS and VLBW

Our fully adjusted models suggested that LFS exposure during the 9 months before birth was significantly associated with an increased risk of LBW or VLBW. According to the estimates, the risks of LBW and VLBW increased by 2.80% (95% CI 0.97–4.66) and 11.68% (95% CI 3.59–20.40), respectively, for each 1 µg/m$^3$ increase in exposure to fire-sourced PM$_{2.5}$. Compared to LBW, VLBW was more strongly associated with fire-sourced PM$_{2.5}$. The estimated association for LBW was sensitive to

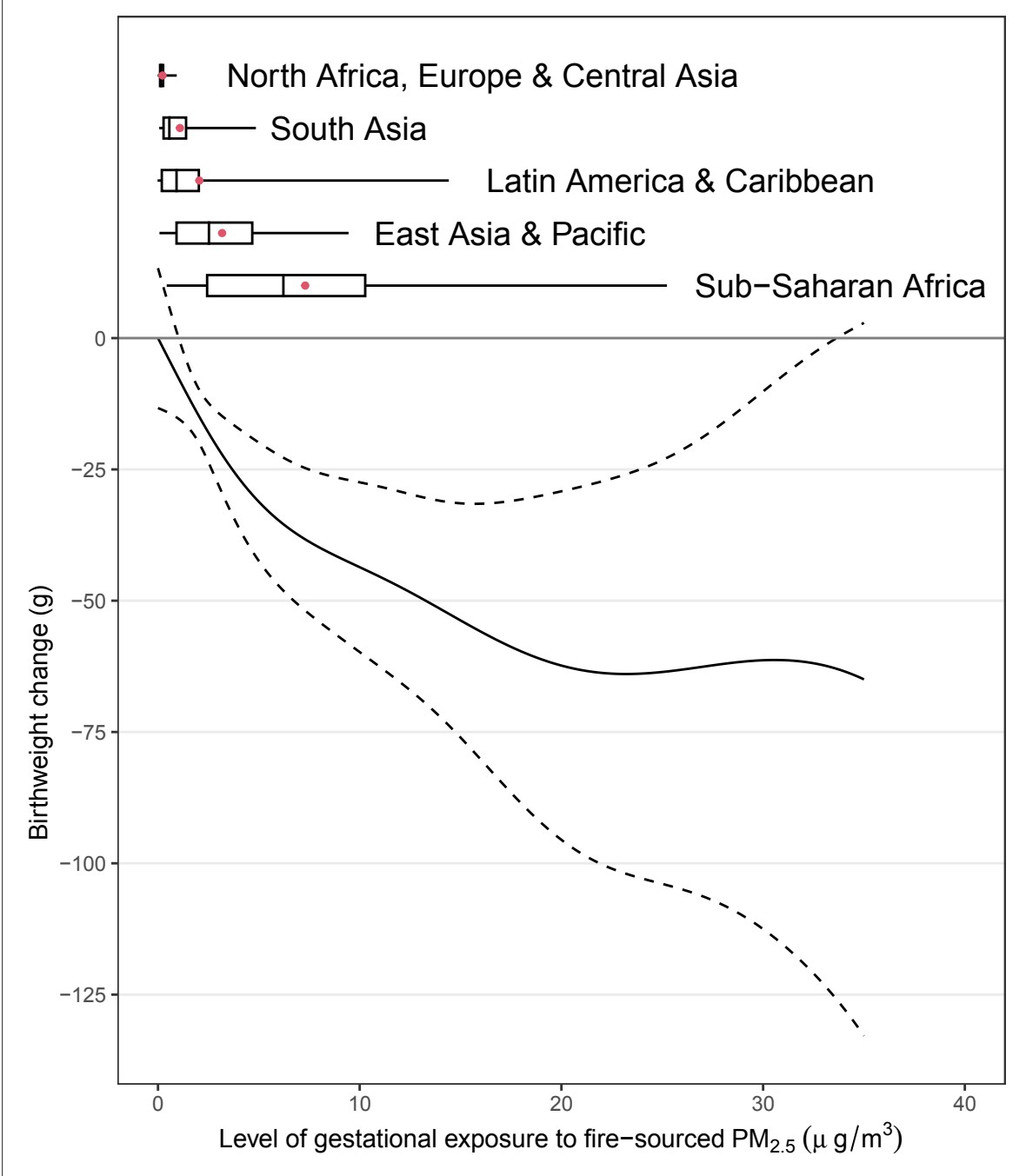

**Figure 3.** The nonlinear association between gestational exposure to fire-sourced $PM_{2.5}$ and change in birthweight. The solid line represents the point estimates, the dashed line represents the 95% confidence intervals, the boxplots represent the distributions of different exposure levels by regions, and the red dots represent the mean exposure level.

different covariate adjustments (*Figure 2*) and was potentially heterogeneous among subpopulations (*Figure 2—figure supplement 3*). The association for VLBW was robust, given the adjustments for different covariates (*Figure 2*), and was less heterogeneous (*Figure 2—figure supplement 3*). Among all effect modifiers, only the type of cooking energy, which was indicated for indoor sources of particulate matter, significantly changed the association between VLBW and fire-sourced $PM_{2.5}$ (p-value = 0.02, *Figure 2—figure supplement 3*). The use of unclear cooking energy, as a competing risk factor for fire-sourced $PM_{2.5}$, significantly weakened the association between VLBW and fire-sourced $PM_{2.5}$.

To further explore why VLBW was more strongly associated with fire-sourced PM$_{2.5}$ than was LBW, we developed a model to estimate the baseline-varying association between birthweight and fire-sourced PM$_{2.5}$. As a result, the absolute change in birthweight for per unit exposure to fire-sourced PM$_{2.5}$ varied with the baseline of family-level mean birthweight (*Figure 4—figure supplement 1*). Livebirths with an extremely low or high baseline birthweight were more susceptible to fire-sourced PM$_{2.5}$ than were those with a moderate baseline birthweight in terms of the absolute birthweight

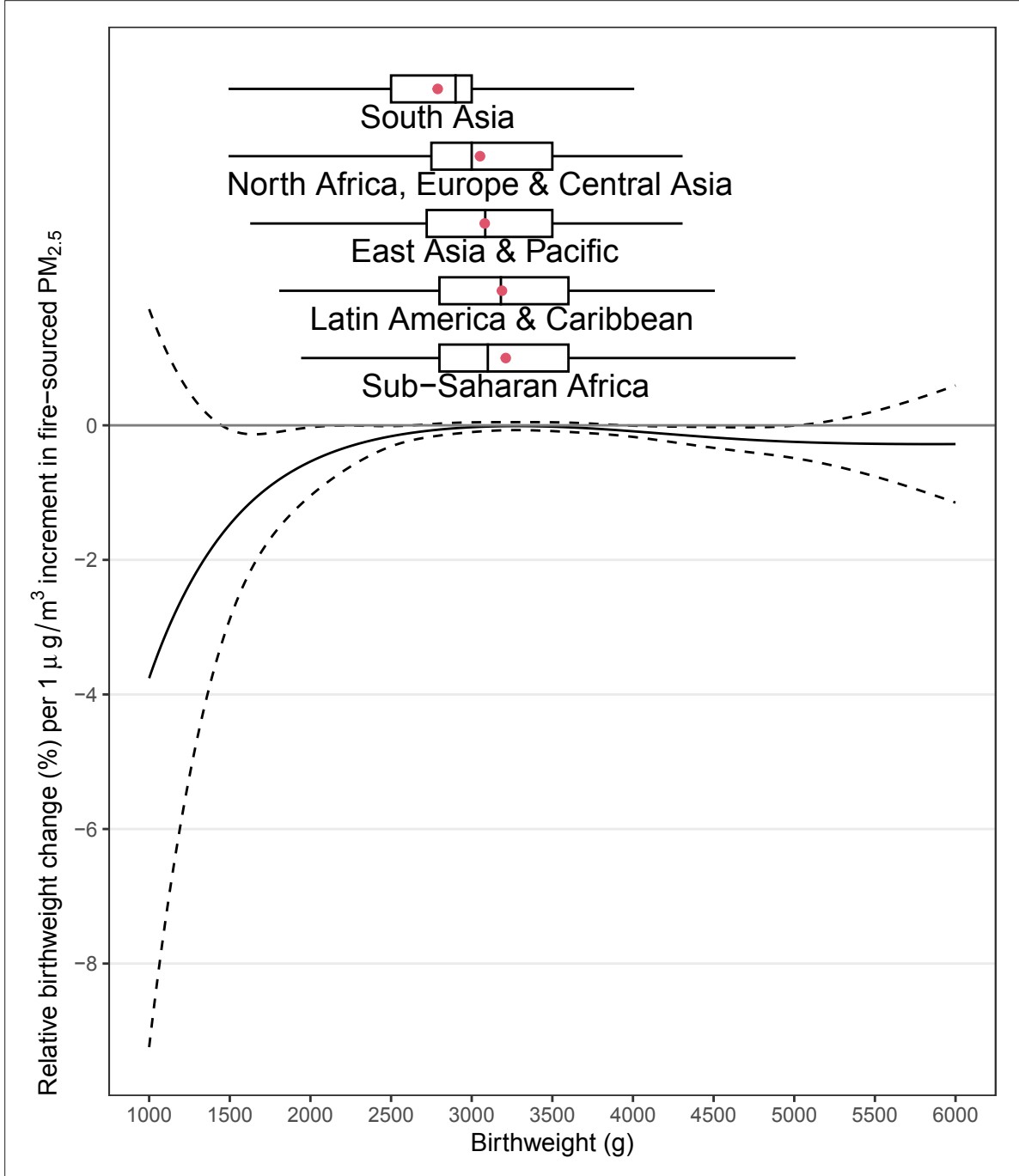

**Figure 4.** Baseline-varying effect of fire-sourced PM$_{2.5}$ on birthweight. The lines show the estimated association (y-axis) between gestational exposure to fire-sourced PM$_{2.5}$ and relative change in birthweight, given different baseline birthweights (x-axis). The solid line represents the point estimates, the dashed line represents the 95% confidence intervals, and the boxplots represent the distributions of different birthweights by region.

The online version of this article includes the following figure supplement(s) for figure 4:

**Figure supplement 1.** The baseline-varying association between gestational exposure to fire-sourced PM$_{2.5}$ and absolute birthweight change.

change (*Figure 4—figure supplement 1*). As a reduction in birthweight and a family-level baseline co-determined the increased incidence of LBW or VLBW, the relative change in birthweight reflected the relevant risk. Newborns from families with a lower baseline were more susceptible to fire-sourced $PM_{2.5}$ in terms of a relative birthweight reduction (*Figure 4*). This finding partially explains why gestational exposure to fire-sourced $PM_{2.5}$ was robustly and strongly associated with VLBW. According to our estimates, a 1 μg/m$^3$ increase in fire-sourced $PM_{2.5}$ was associated with a relative birthweight reduction of 1.47% (95% CI 0.06–2.88), 0.54% (95% CI 0.03–1.05), 0.16% (95% CI 0.01–0.31), or 0.03% (95% CI −0.04–0.10) for livebirths with a baseline birthweight of 1500, 2000, 2500, or 3000 g, respectively.

## Sensitivity analyses

First, we explored potential recall bias using sensitivity analyses, and the estimated associations were not significantly different according to recall period (*Figure 2—figure supplement 3*). Second, we examined the spatial variations in the estimated associations. Although analysis of differences in region-specific associations reported nonsignificant results (*Figure 2—figure supplement 3*), we could not completely rule out the possibility of spatial heterogeneity in the estimated results. As most of our analyzed livebirths were from Sub-Saharan Africa, only those estimates were significant, and the remaining region-specific results had wide-ranging 95% CIs (*Figure 2—figure supplement 3*). Third, we explored whether the estimated adverse effects were attributable to ambient LFS exposure or other impacts (e.g., social stress) relevant to landscape fires. The estimated effects of exposure to fire-sourced $PM_{2.5}$ were not significantly different between locations without burning and those with burning (*Figure 2—figure supplement 3*). These results confirm that ambient exposure was the major exposure pathway for our analyzed outcomes.

## Discussion

This study revealed an association between increased risk of VLBW and exposure to LFS during pregnancy in 54 LIMCs. We found that gestational exposure to fire-sourced $PM_{2.5}$ potentially reduced birthweight, and the infants with a low family-level mean of birthweight were susceptible to the effect of LFS. We present the first sibling-matched study on the association between birthweight and fire-sourced $PM_{2.5}$ from multiple LIMCs.

LFS contains numerous hazardous pollutants and exerts both short- and long-term effects on human health (*Xu et al., 2020*). An adverse birth outcome is one of the consequences. *Holstius et al., 2012* estimated that the mean birthweights of fire-exposed infants are 7.0 g (95% CI 11.8–2.2), 9.7 g (95% CI 14.5–4.8), and 3.3 g (95% CI −0.6–7.2) lower than those of unexposed infants when wildfires occurred during the first, second, and third trimesters, respectively. In a retrospective cohort study in Brazil, fire-exposed mothers (defined as the highest $PM_{2.5}$ concentration quartile) reported a higher risk of LBW than did unexposed mothers. The adjusted odds ratios for exposure during the second and third trimesters were 1.51 (95% CI 1.04–2.17) and 1.50 (95% CI 1.06–2.15), respectively (*Cândido da Silva et al., 2014*). *Abdo et al., 2019* showed that a 1 μg/m$^3$ increase in the wildfire-sourced $PM_{2.5}$ concentration during the first trimester was associated with a 5.7 g (95% CI 0.4–11.1) reduction in birthweight in the United States. In contrast, an Australian cross-sectional study reported that the average birthweight of male infants exposed to the 2003 Canberra wildfires was significantly higher than that of unexposed peers (*O'Donnell and Behie, 2015*). That study attributed the positive association between wildfire smoke and birthweight change to the potentially increased levels of maternal blood glucose after exposure (*O'Donnell and Behie, 2015*). The mixed results might be due to differences among studies in terms of epidemiological design, population characterization, exposure time window, and competing risk factors. Our results are comparable with most previous findings. Additionally, although there are few studies on fire-sourced $PM_{2.5}$ similar to ours, a large number of studies have focused on the effect of gestational exposure to urban ambient $PM_{2.5}$ on birthweight. A recent meta-analysis, which included more than 2.7 million participants, reported that the pooled risk ratio of LBW for a 10 μg/m$^3$ increase in $PM_{2.5}$ exposure was 1.081 (95% CI 1.043–1.120; *Li et al., 2020*).

The family-level mean birthweight varied widely between households (*Table 1*). These variations may have been caused by many factors, including genetics, socioeconomic position (e.g., nutrition), and indoor household pollution. As birthweight is positively correlated with gestation length, a poor

family-level baseline birthweight comprehensively indicates a combined risk of short gestation and LBW. Because the cumulative effect of fire-sourced $PM_{2.5}$ was observed to increase with exposure duration (*Figure 2—figure supplement 2*), a larger birthweight at the family level, as indicated for a longer gestation, increased the absolute effect of the per-unit LFS exposure (*Figure 4—figure supplement 1*).

As our models did not control for gestation length due to the lack of relevant data, the estimated associations reflected the joint outcome of LFS exposure by reducing gestation length or birthweight. The fetus grows in a sublinear temporal pattern (*Salomon et al., 2007*) during the late gestational stage, which suggests its weight increases to an approximately stable level before birth. Considering this sublinear relationship and given the different baseline birthweights, the same reduction in gestation length led to different reductions in birthweight. The marginal effect of a shorter gestation period on the reduction in birthweight is larger for an infant with a lower baseline birthweight. For instance, if an infant is expected to reach the stable level of fetal weight, a marginal reduction in its gestational age will not considerably change the birthweight. As exposure to LFS has been reported to cause a short gestation (*Abdo et al., 2019*), its indirect effect on reduced birthweight can be enhanced by poor baseline maternal health.

Reducing the risk of LBW is one of the 2025 WHO global targets, and it contributes to achieving the Sustainable Development Goals. Although major interventions to reduce LBW have focused on improving maternal nutrition, environmental toxic pollutants, also identified as risk factors, can additionally contribute to the relevant disease burden. For instance, exposure to air pollution was linked to a global burden of 476,000 infant deaths in 2019 during the first month of life via increased risks of LBW and premature birth (*Murray et al., 2020*). LFS induces peak concentrations of $PM_{2.5}$ lasting for a few weeks or months. Its spatial distribution is also different from that of urban $PM_{2.5}$. It is highly clustered in forestry areas (*Figure 1*). Compared to urban $PM_{2.5}$, fire-sourced $PM_{2.5}$ is distributed more skewedly. Therefore, the individual-level impacts on exposed mothers are not negligible. Particularly, LFS exposure frequently occurred in LMICs (*Figure 1*), where the fire-sourced $PM_{2.5}$ adversely affected infants with other risk factors, such as malnutrition. However, due to the lower specificity of our model evaluating birthweight without adjusting for gestation length, we could not evaluate the clinical outcomes or disease burden of the LFS-attributed birthweight reductions. As there is an increasing probability of landscape fire under global climate change, assessing the adverse impacts of LFS exposure is of public health importance to protect maternal health in LMICs. Future studies should be conducted to answer this question.

This study was associated with the following limitations. First, the lack of data on specific gestation lengths in the DHSs limited our exploration. Without this information, we could not distinguish the exposure time window, which introduced potential exposure misclassifications into the epidemiological analyses. Particularly, for the newborns with lower birthweight, their gestations could be shorter, which could lead to a larger likelihood of exposure misclassification. This can explain why the exposures at the 4–6 months before birth were significantly linked to all examined outcomes (*Figure 2—figure supplement 1*). Compared to exposures during the time window, those at longer lags (e.g., lag-8 or -9 month) tend to be pre-pregnancy rather than gestational exposure. Also, we could not determine the clinical importance of the analyzed birthweight without normalizing it to gestation length. Second, characterizing LFS exposure is always challenging. Previous studies measured exposure using surrogates, such as the temporal duration of landscape fires or satellite remote sensing of fire points. To increase interpretability, we utilized fire-sourced $PM_{2.5}$ as a direct indicator of ambient exposure to LFS. However, as the exposure assessments were based on a CTM, the uncertainties introduced by the modeling procedure could not be avoided completely. The potential errors enhanced the exposure misclassifications in this study and thus may have resulted in a biased association. For CTM simulations, their accuracy depends on quality of the inputted emission inventories. Since the large fires can be easily captured by satellite images, which have been considered in the inventory utilized in this study, we believe the effect estimated from a high-exposure region is less biased than that from a low-exposure region. According to the region-specific estimates (*Figure 2—figure supplement 3*), we find that the estimated effects are significant in the highest-exposure region, Sub-Sahara Africa (*Figure 3*), but not in the rest. The result suggests that the exposure misclassification caused and introduced by CTM simulations may lead to underestimated associations. Third, since the analyzed samples were living newborns, our estimates were potentially subjected to survival bias. For instance,

fire-sourced PM$_{2.5}$ was associated with extremely reduced birthweight, which increases the probability of stillbirth or neonatal death. In other words, our sample may have included fewer embryos more susceptible to fire-sourced PM$_{2.5}$ than expected. Therefore, survival bias could have led to an underestimated association by ignoring those susceptible embryos. Fourth, although there have been some previous studies analyzing the birthweight outcome using DHS data (*Bellizzi and Padrini, 2020*), the self-reported records may be inaccurate due to potential recall bias and lead to potential outcome misclassification. The effect of LFS should be confirmed in future by high-quality data on clinical diagnosis of LBW or VLBW. Finally, a sibling-matched study requires a selected sample, which may have lowered the representativeness of this study. The inclusion of infants surveyed by the DHS in this study was determined by many factors, such as family size and planning, willingness to bear a child, socioeconomic position, and contraceptive behavior. However, some of those factors could not be easily measured, which impeded the development of a set of sampling weights to increase the representativeness of our estimates.

In conclusion, we used a global-scale sibling-matched case–control study to identify an association between gestational exposure to fire-sourced PM$_{2.5}$ and reduced birthweight in LMICs. The association was sensitive to regression model settings and potential heterogeneity among different subpopulations. Newborns from families with a low baseline level of birthweight were susceptible to LFS-associated birthweight reduction, which was consistent with our finding of a strong and robust association between fire-sourced PM$_{2.5}$ and VLBW. This study adds to the epidemiological evidence on the adverse effects of LFS on birthweight. Relevant interventions regarding frequently occurring landscape fires, such as climate change mitigations, should be taken to protect maternal and infant health.

## Acknowledgements

This work was supported by National Natural Science Foundation of China (4217050142), PKU-Baidu Fund (2020BD031), the Fundamental Research Funds for the Central Universities (BMU2021YJ042), and the Energy Foundation (G-2107-33169) for TX; and CAMS Innovation Fund for Medical Sciences (2017-I2M-1-004) for TG.

## Additional information

### Funding

| Funder | Grant reference number | Author |
| --- | --- | --- |
| PKU-Baidu Fund | 2020BD031 | Tao Xue |
| Fundamental Research Funds for the Central Universities | BMU2021YJ042 | Tao Xue |
| CAMS Innovation Fund for Medical Sciences | 2017-I2M-1-004 | Tianjia Guan |
| Energy Foundation | G-2107-33169 | Tao Xue |
| National Natural Science Foundation of China | 4217050142 | Tao Xue |

The funders had no role in study design, data collection and interpretation, or the decision to submit the work for publication.

### Author contributions

Jiajianghui Li, Data curation, Formal analysis, Methodology, Writing – original draft; Tianjia Guan, Conceptualization, Funding acquisition, Writing – original draft; Qian Guo, Formal analysis, Investigation, Writing – original draft; Guannan Geng, Data curation, Investigation, Writing - review and editing; Huiyu Wang, Formal analysis, Investigation, Writing - review and editing; Fuyu Guo, Investigation, Writing - review and editing; Jiwei Li, Methodology, Writing - review and editing; Tao Xue, Conceptualization, Data curation, Funding acquisition, Supervision, Writing - review and editing

## Author ORCIDs

Tianjia Guan (iD) http://orcid.org/0000-0002-7820-2898
Qian Guo (iD) http://orcid.org/0000-0003-3343-2849
Huiyu Wang (iD) http://orcid.org/0000-0002-0841-0288
Tao Xue (iD) http://orcid.org/0000-0002-7045-2307

## Ethics

Human subjects: Procedures and questionnaires for standard DHS surveys have been reviewed and approved by ICF Institutional Review Board. All analyses are based on the open-accessed DHS data. The research plan has been approved by DHS, and all analyses adhere the guideline of data usage from DHS.

## Decision letter and Author response

Decision letter https://doi.org/10.7554/69298.sa1
Author response https://doi.org/10.7554/69298.sa2

---

## Additional files

### Supplementary files

• Supplementary file 1. Summary of analyzed variables. (a) Comparison between the $PM_{2.5}$ concentrations simulated by GEOS-Chem model and those estimated from satellite measurements and other inputs. In the comparison, the satellite-based estimates are utilized as the gold-standard referent values and the GEOS-Chem simulations as predictions. (b) Population characteristics.

• Transparent reporting form

• Source code 1. The R codes and data to reproduce *Figures 1–4*.

• Reporting standard 1. Checklist of STROBE items.

### Data availability

All data analysed during this study is included in the manuscript and Supporting Information files. Data is all from publicly available sources. Specifically, the health data can be directly accessed from the Demographic and Health Surveys website, https://www.dhsprogram.com/, after a free registration.

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
