## [Decision Letter]

**Acceptance summary:**

This paper reports the results of an analysis of the association between maternal exposure to landscape fire smoke during pregnancy and low birthweight of the offspring. Given the increasing number, intensity, and duration of landscape fires across the globe as well as the impact of low birthweight on public health, the manuscript will be of interest to both scientists and policymakers. The size of the study population drawn from 54 low and middle-income countries makes the paper an important contribution to the literature on the adverse health effects of biomass fire smoke.

**Decision letter after peer review:**

Thank you for submitting your article "Exposure to landscape fire smoke extremely reduced birthweight in low- and middle-income countries: findings from a siblings-matched case-control study" for consideration by *eLife*. Your article has been reviewed by 1 peer reviewer, and the evaluation has been overseen by Eduardo Franco as the Senior and Reviewing Editor. The following individual involved in review of your submission has agreed to reveal their identity: John Balmes (Reviewer #1).

You will find below the concerns and suggestions that you would have to address to have your submission considered for publication in *eLife*. Please submit a revised version that addresses these concerns directly. Although we expect that you will address these comments in your response letter, we also need to see the corresponding revision clearly marked in the text of the manuscript. Some of the comments may seem to be simple queries or challenges that do not prompt revisions to the text. Please keep in mind, however, that readers may have the same perspective as the reviewers. Therefore, it is essential that you attempt to amend or expand the text to clarify the narrative accordingly.

General Comments:

Li and colleagues used data from 2000 to 2014 in 54 low and middle-income countries (LMICs) to study the association between exposure to landscape fire smoke PM2.5 and birthweight, including very low birthweight. While there is a relatively robust epidemiological literature that supports an association between non-biomass fire smoke PM2.5 and low birthweight, there are relatively few studies that are specific to biomass smoke PM2.5 and birthweight. The authors of this paper conducted their study to specifically address this data gap. They took advantage of satellite data which provide estimates of PM2.5 levels that are now available for most locations in the world at a high geographic resolution (0.5 x 0.5 km). They enhanced the satellite exposure data using a chemical transport model to distinguish fire-sourced PM2.5 from non-fire PM2.5. The exposure modeling approach is sophisticated as is the statistical analysis of the association between the fire-sourced PM2.5 exposure estimates and birthweight outcomes.

The study has multiple strengths, including the first study of the association between fire-sourced PM2.5 and birthweight to use a sibling-matched case-control design, a large sample size (227,948 births born to 109,137 mothers), the focus on LMICs, the exposure modeling, a careful statistical analytic approach with alternate non-linear regression and sensitivity analyses, and the outcome of very low birthweight that is one of the World Health Organization targets to reduce the global burden of disease. Limitations notwithstanding, this is an impactful study. The results of the authors' analyses provide strong support for the concept that exposure to biomass smoke -- whether from a landscape fire set by farmers, a wildfire, or cooking with solid fuels -- can lead to low birthweight. This concept is especially important for LMICs that have large portions of their populations engaging in slash and burn agriculture and/or cooking with solid fuels. Given that reducing the incidence of low birthweight is a necessary to meet the 2025 United Nations Sustainable Development Goals, it is incumbent that policies to reduce landscape fires and household air pollution from cooking with solid fuels be considered by governments of LMICs. Such policies would also have a climate change mitigation benefit through reduction of greenhouse gases and aerosols.

Future research efforts to actually measure landscape fire smoke PM2.5 in different locations to provide ground-truthing for the chemical transport model exposure estimates used by the authors would be useful as would a study that could obtain gestation duration data.

Essential revisions:

Despite these strengths there several weaknesses of the study. Perhaps the most important is potential misclassification of exposure to landscape fire PM2.5 because levels of this pollutant were not measured, but rather estimated using a chemical transport model.

Another important weakness is the lack of gestation duration data that limited the primary analysis to a typical 9-month exposure period which also adds to exposure misclassification because some low-birthweight babies have shorter gestation periods.

A third weakness is that birthweight was not measured but rather provided by maternal recall which may not always be reliable, especially in rural areas of low-resourced countries when children are born at home. The authors acknowledge the potential for misclassification of exposure to fire-sourced PM2.5, but not for the birthweight outcome.

Title I suggest changing the title to "Exposure to landscape fire smoke extremely and reduced birthweight in low- and middle-income countries".

Line 46 I suggest the concluding statement of the Abstract be revised as follows: "Our findings indicate that gestational exposure to LFS harms fetal health."

Line 68 "Rarely discussed" is too strong. As the authors note in the Discussion, there are other published studies that have addressed the effect of biomass smoke exposure on birthweight.

Line 69 Should be "LFS" rather than LPS.

Line 85 Should be "adverse birth outcomes" rather than birth defects.

Line 90 Again, "little is known about fire-sourced PM2.5" is too strong given the published literature.

Line 151 Shouldn't this be the 9-month average preceding the "birthdate" rather than survey month?

Line 225 Here and elsewhere in the manuscript, I suggest "live births" rather than samples.

Line 358 It is not clear to this reviewer that "the association of birthweight with fire-sourced PM2.5 should be stronger than that with urban PM2.5." While this may turn out to be true with future research findings, without any data, this is purely speculative.

Line 386 Exposures to urban PM2.5 are also differentially distributed with low-income minority populations suffering higher exposures.

---

## [Author Response]

Essential revisions:Despite these strengths there several weaknesses of the study. Perhaps the most important is potential misclassification of exposure to landscape fire PM2.5 because levels of this pollutant were not measured, but rather estimated using a chemical transport model.

We agree with the reviewer’s opinion that using a chemical transport mode to calculate landscape fire PM_2.5_ may introduce potential misclassification. According to this comment, we make the following two revisions.

1. We enhance our statements on evaluation of the CTM model by added a new paragraph, which reads “Since fire-sourced PM_2.5_ is mixed with particles from other sources, it cannot be easily measured. Although the CTM-based approach has been widely utilized in a few large-scale studies (Xue, Geng, Li, et al., 2021; Ye et al., 2021) to assess the LFS exposure, accuracy of its results cannot be evaluated directly. Therefore, in this study, we assessed the overall performance of the GEOS-Chem model by comparing the simulated PM_2.5_ concentrations with the satellite-based estimates (Van Donkelaar et al., 2016). At the surveyed locations (i.e., the grey dots in Figure 1), we found the two estimators were in good agreement with each other (Pearson correlation coefficient, R^2^ = 0.76; more details on the comparison are shown in Supplementary file 1b).”. Compared to emission inventories inputted into the GEOS-Chem model, the one on landscape fires can be in good quality, because the fire emission inventory is constrained by satellite image of burned areas. Our evaluation shows the GEOS-Chem model can well mimic the chemical and physical processes in PM_2.5_ from all sources. If this also holds for fire-sourced PM_2.5_, it is reasonable to believe the model can estimate the fire-related exposure well, given the high-quality of inputted emission inventory. Therefore, this approach has been widely used in recent studies (e.g., Ye et al., Chen et al., Xue et al., the Lancet Planetary Health 2021).

2. We enhance the discussions on the limitations relevant to the exposure misclassifications. The modified sentences now read as “Second, characterizing LFS exposure is always challenging. Previous studies measured exposure using surrogates, such as the temporal duration of landscape fires or satellite remote sensing of fire points. To increase interpretability, we utilized fire-sourced PM_2.5_ as a direct indicator of ambient exposure to LFS. However, as the exposure assessments were based on a CTM, the uncertainties introduced by the modeling procedure could not be avoided completely. The potential errors enhanced the exposure misclassifications in this study and thus may have resulted in a biased association. For CTM simulations, their accuracy depends on quality of the inputted emission inventories. Since the large fires can be easily captured by satellite images, which have been considered in the inventory utilized in this study, we believe the effect estimated from a high-exposure region is less biased than that from a low-exposure region. According to the region-specific estimates (Figure 2—figure supplement 3), we find the estimated effects are significant in the highest-exposure region, Sub-Sahara Africa (Figure 3), but not in the rest. The result suggests the exposure misclassification caused introduced by CTM simulations may lead to underestimated associations.”.

Another important weakness is the lack of gestation duration data that limited the primary analysis to a typical 9-month exposure period which also adds to exposure misclassification because some low-birthweight babies have shorter gestation periods.

Thank you for the reminder. We agree that using a typical 9-month exposure may add some misclassification, particularly for the low-birthweight samples. According to the comment, we make the following revisions.

1. We enhance discussions on the relevant limitation. The modified sentences now read as “First, the lack of data on specific gestation lengths in the DHSs limited our exploration. Without this information, we could not distinguish the exposure time window, which introduced potential exposure misclassifications into the epidemiological analyses. Particularly, for the newborns with lower birthweight, their gestations could be shorter, which could lead to a larger likelihood of exposure misclassification. This can explain why the exposures at the 4-6 months before birth were significantly linked to all examined outcomes (Figure 2—figure supplement 1). Compared to exposures during the time-window, those at longer lags (e.g., lag-8 or -9 month) tend to be pre-pregnancy rather than gestational exposure. Also, we could not determine the clinical importance of the analyzed birthweight without normalizing it to gestation length.”.

2. We enhance our statement on the sensitivity analysis of different exposure time-windows. The modified sentences not read as “In sensitivity analyses, we first derived the lag-distributed model to explore how the association varied during the exposure time window. As the specific duration of gestation was not recorded, before actual data analysis, we utilized the 9-month average preceding the birth month as the major exposure time-window. After conducting the major analysis, we re-estimated the effects of fire-sourced PM_2.5_ on birthweight change, low birthweight or very low birthweight, by different time-windows using the lag-distributed model.”. The relevant results have been discussed in our Results section. The key results are presented in Figure 2—figure supplement 1.

A third weakness is that birthweight was not measured but rather provided by maternal recall which may not always be reliable, especially in rural areas of low-resourced countries when children are born at home. The authors acknowledge the potential for misclassification of exposure to fire-sourced PM2.5, but not for the birthweight outcome.

Suggestion accepted. According to the comments, we make the following revisions.

1. We enhance our statements on the sensitivity analysis on recall bias. To illustrate that, in method section, we add a few words, which read as “Fourth, to examine potential recall bias, we estimated the associations by strata of the durations from birth to survey time (the duration is defined as recall period in this study). We assumed a shorter recall period indicated for a lower likelihood of outcome misclassification.”. The relevant result is illustrated using the following words, which read as “First, we explored potential recall bias using sensitivity analyses, and the estimated associations were not significantly different according to recall period (Figure 2—figure supplement 3)”. Actually, we find the weakest association for either low birthweight or very low birthweight is observed for the subgroup of newborns closest to the survey date. This is may be caused by the small size of subgroup samples. It is possible, because mothers just after childbirth may not like to participate the household survey due to non-optimal physiological and psychological conditions.

We add the recall bias and the relevant outcome misclassification as a new limitation. To state that, in Discussion section, we add a few words, which read as “Fourth, although there have been some previous studies analyzing the birthweight outcome using DHS data (Bellizzi and Padrini, 2020), the self-reported records may be inaccurate due to potential recall bias and lead to potential outcome misclassification. The effect of LFS should be confirmed in future by high-quality data on clinical diagnosis of LBW or VLBW.”.

Title I suggest changing the title to "Exposure to landscape fire smoke extremely and reduced birthweight in low- and middle-income countries".

The reviewer’s comment reminds us that the “extremely” leads to ambiguity. In previous title, we aim to highlight the association between landscape fire smoke and very low birthweight, by using the word. To avoid the ambiguity, we revise the title as “Exposure to landscape fire smoke reduced birthweight in low- and middle-income countries: findings from a siblings-matched case-control study”. If the reviewer has further suggestion on the title, we would like to make more corrections.

Line 46 I suggest the concluding statement of the Abstract be revised as follows: "Our findings indicate that gestational exposure to LFS harms fetal health."

Suggestion accepted. We modified our abstract as “Our findings indicate that gestational exposure to LFS harms fetal health.”

Line 68 "Rarely discussed" is too strong. As the authors note in the Discussion, there are other published studies that have addressed the effect of biomass smoke exposure on birthweight.

Suggestion accepted. We modified our manuscript as:

“However, the health impacts of LFS on susceptible pregnant women, another highly vulnerable group due to gestation-related physiological changes, such as an increased breathing rate during pregnancy, are not thoroughly discussed.”

Line 69 Should be "LFS" rather than LPS.

We apologize for making a spelling mistake. We have modified the corresponding part as follows:

“Previous studies have shown that gestational exposure to air pollutants, including those derived from LFS and other emissions”

Line 85 Should be "adverse birth outcomes" rather than birth defects.

Suggestion accepted. We have modified our manuscript as:

“All risk factors for adverse birth outcomes should be screened thoroughly to protect maternal and infant health.”

Line 90 Again, "little is known about fire-sourced PM2.5" is too strong given the published literature.

Suggestion accepted. We have modified the sentence as:

“Although a growing number of studies have suggested the deleterious effects of urban air pollution on LBW (C. Li et al., 2020; X. Y. Li et al., 2017), the exposure-response function specifically for fire-sourced PM_2.5_ has been insufficiently studied.”.

Line 151 Shouldn't this be the 9-month average preceding the "birthdate" rather than survey month?

We apologize for making a mistake and not expressing our meaning clearly. We have modified the manuscript as:

“As the DHSs did not record the specific duration of gestation, we utilized the 9-month average preceding the birth month as exposure time during pregnancy.”

Line 225 Here and elsewhere in the manuscript, I suggest "live births" rather than samples.

Suggestion accepted. We have replaced all the “samples” with “livebirths” in our manuscript.

Line 358 It is not clear to this reviewer that "the association of birthweight with fire-sourced PM2.5 should be stronger than that with urban PM2.5." While this may turn out to be true with future research findings, without any data, this is purely speculative.

Suggestion accepted. We remove this sentence in the revised manuscript.

Line 386 Exposures to urban PM2.5 are also differentially distributed with low-income minority populations suffering higher exposures.

We agree with the reviewer, and revise the relevant sentences correspondingly. We don’t mean that the urban PM_2.5_ exposure is evenly distributed. Accordingly, now the relevant sentences are revised as:

“LFS induces peak concentrations of PM_2.5_ lasting for a few weeks or months. Its spatial distribution is also different from that of urban PM_2.5_. It is highly clustered in forestry areas (Figure 1). Compared to urban PM_2.5_, fire-sourced PM_2.5_ is distributed more skewedly.”